# Effects of Dwarfing Interstock Length on the Growth and Fruit of Apple Tree

Shasha Zhou [†], Zhen Shen [†], Baoying Yin, Bowen Liang, Zhongyong Li, Xueying Zhang * and Jizhong Xu *

College of Horticulture, Hebei Agricultural University, Baoding 071001, China
* Correspondence: zhangxueying1996@163.com (X.Z.); xjzhxw@126.com (J.X.)
† These authors contributed equally to this work.

**Abstract:** There is no report on the effect of the length of Jizhen 2 interstock on the growth and fruit quality of Tianhong 2 apple trees, which are usually grown in Baoding, Hebei Province, China. We surveyed the tree size, branch types, fruit set, fruit quality and root parameters of 3–5-year-old 'Tianhong 2/Jizhen 2/*Malus × robusta* Rehder' apple trees, to study the effects of dwarfing interstock length on the growth of the tree's aboveground parts and roots, as well as fruit yield and quality. The tree height and the stem girths of the interstock and scion decreased as interstock length increased, and the dwarfing degree of the apple trees gradually increased. Trees with an interstock length of 30 cm had the fewest long branches, the most short branches, and the greatest proportion of short branches. An interstock length of 30 cm provided the highest fruit-set rate, the highest yield per tree and per unit cross-sectional area, the highest single fruit weight, the highest soluble: acid ratio, the highest color brightness (L*), and better red skin coloration (higher a*) of the fruit skin. The root length density, root surface area density, and root volume density exhibited two growth peaks in a year, during the slow growth period when spring and autumn shoots are stopped. Root length density, root surface area density, and root volume density decreased with interstock length. Root death peaked during the growth peak period of the autumn shoots, and root length density of dead roots and root turnover frequency increased with the interstock length. A 30 cm length was the most suitable for the Tianhong 2 apple trees when the Jizhen 2 was used as the interstock.

**Keywords:** apple; interstock length; growth; fruit quality; root

## 1. Introduction

Apple dwarfing and dense planting has the advantages of early blooming and fruiting, high yield, and good quality, which has facilitated intensive cultivation management [1], and has become the mainstream and direction of modern apple production globally. Dwarfing interstock is a dwarfing and dense planting technology available for apple, pear, and pomegranate trees, among others [2–4].

Modern high-density orchard systems aim for high yields at an early tree age and vegetative growth control [5]. Thus, growers require dwarfed and well-branched maiden trees for planting intensive orchards. Scions can be dwarfed by grafting onto dwarfing rootstocks [6]. More recently, interstocks have been evaluated on apple [6] and other fruit trees [7,8] to control the vigor of the scion. Previous studies have shown that different dwarfing interstocks [9,10] and dwarfing interstock lengths [11,12] significantly affect plant size, fruit weight, the time of fruit maturation, and increased fruit yield efficiency per plant, providing an opportunity to increase tree density and improve canopy management. Some studies have shown that the dwarfing effect on apple trees increases with the interstock length [6,13], while another study found that different interstock lengths significantly affect the fruit quality and yield of the 'Royal Gala' and 'Nagafu No.2' apples [14].

The interactions between the roots and the aboveground parts are of major interest in the fruit industry due to grafting. The root is the foundation of the growth of fruit trees, as

roots hold plants in place, absorb water and nutrients, and synthesize important substances. The development of the roots and aboveground parts are mutually influential [15]. The nutrients needed for root growth mainly originate from leaf photosynthesis, and the roots provide most of the nutrients needed for the growth of aboveground tissues. Previous studies on the dwarfing interstock length have mainly focused on the final stages of tree growth; however, systematic research on the effect of the dwarfing interstock length on roots growth is lacking.

Jizhen 2 is an apple dwarfing rootstock cultivar selected by Hebei Agricultural University. In 2016, it passed the forest species examination and was approved by Hebei Province [1]. Tianhong 2 (*Malus domestica* Borkh. v. Tianhong 2) apple is a short-branch variety of Fuji (*Malus domestica* Borkh. cv. Fuji) bred by the Apple Research Group of Hebei Agricultural University in 1994 and certified as a new variety by the Forest Tree Variety Approval Committee of Hebei Province, China in 2005.

In south central Hebei Province, *Malus* × *robusta* Rehder is usually used as the rootstock for apples due to its resistance and adaptability, but *M. robusta* is not a dwarf cultivar. Interstocks with dwarfing characteristics are usually used to achieve dwarfing and dense planting. When Jizhen 2 is used as the interstock, trees grafted with Tianhong 2 as the scion show dwarfing, good grafting compatibility, and consistent traits [16]. However, no study has assessed the effects of Jizhen 2 interstock length on the growth, yield and fruit quality of apple trees. Therefore, this study used 3–5–years–old 'Tianhong 2/Jizhen 2/*Malus* × *robusta* Rehder' apple trees as materials to analyze the effects of the dwarfing interstock length on the growth of the aboveground plant parts and roots, as well as the fruit quality, of apple trees. This work aimed to determine the optimal interstock length of Jizhen 2 for Tianhong 2 apples, and provides a theoretical and practical basis for promoting the intensive cultivation mode of dwarfing interstock.

## 2. Materials and Methods

This research was conducted at the Nanshennan Apple Experimental Base, Shunping County, Hebei Province (38°59′ N, 114°54′ E). The area is 291 m above sea level, the average annual sunshine hours are 2500–2600 h, the average annual precipitation is 580 mm, and the frost-free period is about 195 days. The soil is shallow sandy loam soil. *M. robusta* Rehder annual seedlings were used as the rootstocks, and Jizhen 2 was used as the interstock. The Tianhong 2 apple buds were grafted onto the interstocks of different lengths in 2017. The 'Tianhong 2/Jizhen 2/*Malus* × *robusta* Rehder' grafted apple trees were planted in 2017, and the spacing within and between rows was 1.5 m and 4 m, respectively. There were five treatments, including T-10, T-20, T-30, T-40, and T-50, with interstocks lengths of $10 \pm 2$, $20 \pm 2$, $30 \pm 2$, $40 \pm 2$, and $50 \pm 2$ cm, respectively. Each treatment included five replicates; each replicate included one tree, and the trees were managed routinely. The trees were pruned between February and March every year. The pruning intensity was about 20% of the total branches. The tree structure was slender and spindle–shaped. The survey was conducted from 2019 to 2021.

### 2.1. Tree Size and Branch Types

After vegetative growth had stopped in October 2019–2020, the tree height (cm) (from the grafting union above the interstock to the top of the tree), trunk circumference (cm), length of new peripheral shoots (cm)(10 shoots per tree),and crown diameter (cm) (maximum distance of the branches spreading in the east–west direction and the north–south direction) of the apple trees were measured with a flexible ruler.

After the apple trees had entered the leaf–fall period in 2020, the number of different branch types (long, medium, and short branches) was investigated, and the share of different branches in the total number of branches was calculated. For the classification criteria for the branches, short branches were 1.1–5.0 cm, medium branches were 5.1–15.0 cm, and long branches were >15.1 cm.

### 2.2. Determination of Fruit Setand Yield

Five representative branches (main branch directly attached to the central stem, 3–4 years old, 1.4–1.6 m high from the ground to the canopy) from each tree were surveyed for the numbers of inflorescences and flowers during the 2021 full-bloom period. After one month, from late May to early June, in the expansion stage of young fruit after the period of physiological fruit drop, when the fruit size was 1–1.5 cm in diameter, the number of inflorescences with fruit and the number of fruitlets was counted. The fruit–set rates of the inflorescences and of a single flower were calculated as:

Fruit-set rate of the inflorescence (%) = (the number of inflorescences with fruit/the number of inflorescences) × 100%

Fruit-set rate of a single flower (%) = (the number of fruits/the number of flowers) × 100%

In mid-October 2020 and 2021, the number of fruits on each tree at the ripening stage (edible maturity, showing characteristic color, aroma, and flavor, with crisp pulp) was measured. The yield (kg) per plant, and the yield (kg) per $cm^2$ of trunk cross-sectional area (TCSA) (the ratio of yield per plant to cross-sectional area per square centimeter) were determined. The cross-sectional area 10 cm above the grafting interface between the interstock and the variety represented the cross-sectional area of the variety; the cross-sectional area 10 cm below the grafting interface between the interstock and the variety, represented the cross-sectional area of the interstock.

### 2.3. Effect on Fruit Quality

Fifty disease and insect–free uniform fruits (10 fruits per tree) were picked 1.4–1.6 m above the ground and from all sides of the canopy. They were randomly selected from each treatment to measure the following parameters:

(1) Single fruit weight (g): single fruit weight was measured using an electronic balance, and the average value was calculated.

(2) Fruit shape index: the transverse and vertical diameters (mm) of the fruits were measured using digital electronic Vernier calipers. The fruit shape index was the ratio of vertical diameter to transverse diameter.

(3) Fruit flesh firmness ($kg/cm^2$): four sides of the fruit were peeled, and fruit flesh firmness was measured with a GY-4 handheld digital fruit firmness tester.

(4) Malic acid content (%): the apple pulp was cut into small pieces and squeezed. Then, 0.3 mL of the juice was poured into 50 mL of distilled water, and the acidity was measured. Thirty measurements were performed for each treatment. Malic acid content was measured with a GMK-835F pH tester.

(5) Total soluble solids (TSS) content (%): the total soluble solids content was measured with a PAL-1 digital Brix meter (Atago Refractometer, Tokyo, Japan).

(6) The soluble solids: acid ratio: the ratio of total soluble solids to malic acid content.

(7) Chromatic aberration: chromatic aberration was measured with the Mitutoyo CR-400 colorimeter. L* represented the color brightness, a* represented the peel's red–green value, and b* represented the yellow–blue value. After the instrument was calibrated, the fruit was placed under the colorimeter probe, and the measurements were carried out on the front, back, left, and right sides of the fruit.

### 2.4. Calculation of Root Parameters

From March to September 2021, a root scanner was used to detect the growth dynamics of the apple tree roots. A minirhizotron was inserted 60 cm away from the trunk, about 60 cm deep, at a 90° angle to the ground. Root Analysis software was used to process, label, and describe the roots. Data on root length density, root surface density, root volume density, and length density of dead roots were collected [17].

Relevant indexes were calculated based on the unit soil volume ($S \times D$), where $S$ is the soil area measured with the single minirhizotron and $D$ is the soil layer thickness

measured with the minirhizotron. In this experiment, $S = 7\pi \times 85$ cm$^2$ and $D = 0.25$ cm. The root length density (mm·cm$^{-3}$) = $L/(S \times D)$, and L (mm) is the total root length of a single minirhizotron. The root surface density (mm$^2$·cm$^{-3}$) = $SA/(S \times D)$, where $SA$ is the total root surface area of a single minirhizotron (mm$^2$). The root volume density $V$ (mm$^3$·cm$^3$) = $V/(S \times D)$, where $V$ is the total root volume of a single minirhizotron (mm$^3$). When calculating the root length density of dead roots, $L$ is the total length of the dead root in the interval between two image acquisitions. The root system turnover rate was the ratio of the number of annual root death to the number of annual average survivals of fine roots [18].

## *2.5. Data Analysis*

Microsoft Excel 2010 software (Microsoft Inc., Redmond, WA, USA) was used to analyze the data and draw the figures. SPSS 21.0 software (SPSS Inc., Chicago, IL, USA) was used for one-way ANOVA, Duncan's test ($p< 0.05$) was used to detect differences between the groups, and a $p$-value <0.05 was considered significant.

## 3. Results
### *3.1. Effects of Dwarfing Interstock Length on the Vegetative Growth of Apple Trees*
#### 3.1.1. Dwarfing Interstock Lengths Effects on Tree Size

The effects of the dwarfing interstock length on the growth of apple trees for two consecutive years (2019 and 2020) are shown in Tables 1 and 2. In 2019, as interstock length increased, the tree height and the trunk circumference of the interstock and scion decreased. The height of T-10 tree was significantly greater than that of T-50. The T-10 interstock stem girth in 2019 was significantly greater than that of other treatments, while the T-10 scion stem girth was significantly greater than that of T-40 and T-50. The T-10 tree height in 2020 was 378.00 cm, which was significantly greater than that of T-40 and T-50. The T-10 scion stem girth was significantly greater than that of the T-50. In 2019, there was no significant difference in the shoot length of T-10, T-20, or T-50, but the shoot length of T-10 was significantly shorter than that of T-30 and T-40. In 2020, there was no significant difference in the shoot length among the five treatments. The crown diameter in the T-10 rows was significantly greater than that of T-40 and T-50, and the crown diameter between the rows was significantly higher than that of T-50. The tree height gradually decreased with the increase in the dwarfing interstock length.

**Table 1.** Effects of dwarfing interstock length on apple tree growth (2019).

| Treatment | Tree Height cm | Trunk Circumference cm | | Shoot Length cm |
| --- | --- | --- | --- | --- |
| | | Interstock | Scion | |
| T-10 | 234.00 ± 16.26 a | 15.17 ± 3.54 a | 12.91 ± 1.86 a | 39.56 ± 14.5 b |
| T-20 | 222.40 ± 6.91 ab | 12.46 ± 1.19 b | 11.90 ± 0.98 ab | 52.60 ± 11.68 ab |
| T-30 | 220.80 ± 47.52 ab | 12.17 ± 1.22 b | 11.33 ± 1.86 ab | 66.32 ± 11.85 a |
| T-40 | 201.20 ± 17.71 ab | 11.82 ± 1.95 b | 10.33 ± 1.32 b | 61.34 ± 9.64 a |
| T-50 | 189.00 ± 11.42 b | 11.56 ± 0.98 b | 10.45 ± 1.09 b | 41.60 ± 16.98 b |

Note: Lowercase letters indicate significant differences between treatments ($p < 0.05$).

**Table 2.** Effects of different dwarfing interstock lengths on the growth of apple trees (2020).

| Treatment | Tree Height cm | Stem Girth cm | | Shoot Length cm | Crown Diameter cm | |
|---|---|---|---|---|---|---|
| | | Interstock | Scion | | in the Rows | between the Rows |
| T-10 | 378.00 ± 25.24 a | 27.00 ± 2.29 a | 21.33 ± 0.58 a | 23.94 ± 5.83 a | 184.70 ± 13.61 a | 183.30 ± 15.28 a |
| T-20 | 363.33 ± 8.33 ab | 23.20 ± 0.72 b | 20.00 ± 0.5 ab | 19.91 ± 3.67 a | 165.30 ± 5.03 ab | 176.30 ± 3.22 ab |
| T-30 | 357.67 ± 12.66 ab | 21.00 ± 3.61 bc | 19.00 ± 2.65 ab | 19.86 ± 1.81 a | 164.00 ± 11.53 ab | 160.70 ± 16.01 ab |
| T-40 | 342.00 ± 13.86 bc | 19.40 ± 1.51 bc | 18.43 ± 1.60 ab | 19.63 ± 2.02 a | 150.00 ± 17.32 b | 156.70 ± 7.64 ab |
| T-50 | 327.00 ± 4.58 c | 17.40 ± 0.53 c | 17.83 ± 1.26 b | 21.98 ± 3.73 a | 148.30 ± 15.28 b | 153.30 ± 20.82 b |

Note: Lowercase letters indicate significant differences between treatments ($p < 0.05$).

### 3.1.2. Effects of Interstock Length on the Branch Type

The effects of different dwarfing interstock lengths on the apple tree branch type in 2020 are shown in Table 3. There were significant differences in the number of long branches among treatments. T-30 had significantly fewer long branches compared to other treatments. There was no significant difference in the number of medium branches among treatments. The T-30 treatment had significantly more short branches (77.19%) than the other treatments.

**Table 3.** Effects of different dwarfing interstock lengths on the number and share of different types of branches (2020).

| Treatment | Number of Long Branches | Share of Long Branch (%) | Number of Medium Branches | Share of Medium Branches (%) | Number of Short Branches | Share of Short Branches (%) |
|---|---|---|---|---|---|---|
| T-10 | 47.67 ± 7.09 a | 27.41 ± 1.95 a | 17.33 ± 1.53 a | 8.37 ± 1.80 a | 106.67 ± 1.15 c | 64.22 ± 2.25 b |
| T-20 | 32.00 ± 1.00 b | 21.25 ± 4.33 c | 16.67 ± 3.21 a | 9.80 ± 1.62 a | 124.33 ± 4.04 b | 68.95 ± 5.47 b |
| T-30 | 28.00 ± 2.00 c | 14.43 ± 0.62 c | 18.67 ± 2.52 a | 8.38 ± 1.83 a | 149.00 ± 9.64 a | 77.19 ± 6.02 a |
| T-40 | 42.33 ± 4.51 ab | 23.32 ± 1.71 b | 15.67 ± 4.04 a | 8.59 ± 1.17 a | 123.33 ± 2.52 b | 68.09 ± 3.62 b |
| T-50 | 52.33 ± 6.43 a | 25.42 ± 1.99 ab | 21.67 ± 4.93 a | 8.53 ± 0.91 a | 123.67 ± 11.06 b | 66.05 ± 2.36 b |

Note: Lowercase letters indicate significant differences between treatments ($p < 0.05$).

### 3.2. Effects of Dwarfing Interstock Length on Flowering and Fruit-Setting in Apple Trees

Table 4 shows the effects of different interstock lengths on flowering and fruit set. The T-10 treatment had the greatest number of flowers per tree, which was significantly greater than that of T-40 and T-50. The T-30 inflorescences had the highest fruit set rate (87%), followed by T-40 (86%), and that of T-10 was lowest (77%). The fruit-setting rate of a single T-30 flower was significantly higher (34%) than the other treatments.

**Table 4.** Effects of dwarfing interstock length on flowering, and fruit-setting rates (2021).

| Treatment | Flowers per Tree | Fruit Setting Rate of the Inflorescence (%) | Fruit Setting Rate of a Single Flower (%) |
|---|---|---|---|
| T-10 | 143.00 ± 13.25 a | 77.00 ± 8.23 b | 27.00 ± 2.21 b |
| T-20 | 141.25 ± 11.12 a | 84.00 ± 6.07 ab | 27.00 ± 1.56 b |
| T-30 | 119.75 ± 17.43 ab | 87.00 ± 2.01 a | 34.00 ± 5.09 a |
| T-40 | 105.00 ± 10.10 b | 86.00 ± 5.22 a | 31.00 ± 4.60 b |
| T-50 | 110.00 ± 15.69 b | 84.00 ± 3.34 ab | 25.00 ± 5.66 b |

Note: Lowercase letters indicate significant differences between different treatments ($p < 0.05$).

### 3.3. Impact on Yield

The 2020 and 2021 results showed significant differences in the yield per plant and per unit cross-sectional area (TCSA) among the different interstock length treatments. As shown in Table 5, T-20 had the highest yield per plant in 2020, with 3909.97 g, which was significantly higher than T-40; however, there was no significant difference between T-20 and T-30. In 2020, the yield per cm$^2$ TCSA of the T-50interstock was the highest (0.12 kg/cm$^2$), which was significantly higher than that of T-10 and T-40.The yield per TCSA of the T-30 scion was the highest (0.13 kg/cm$^2$), which was significantly higher than T-10 and

T-40. In 2021, the yield per T-20 plant was the highest (16,867.90 g), which was significantly higher than T-10, T-40, and T-50. The yield per TCSA of T-30 interstock was the highest (0.25 kg/cm$^2$), which was significantly higher than T-10 and T-40. The yield per TCSA of the T-30 scion was the highest (0.23 kg/cm$^2$), which was significantly higher than T-10 and T-50.

**Table 5.** Effects of different dwarfing interstock lengths on yield (2020 and 2021).

| Treatment | Yield per Tree (kg) 2020 | Yield per Tree (kg) 2021 | Yield per Unit Cross-Sectional Area (kg/cm$^2$) | | | |
| | | | According to Interstock 2020 | According to Scion 2020 | According to Interstock 2021 | According to Scion 2021 |
|---|---|---|---|---|---|---|
| T-10 | 2.93 ± 0.56 ab | 8.93 ± 0.49 b | 0.05 ± 0.004 b | 0.08 ± 0.007 b | 0.12 ± 0.007 b | 0.15 ± 0.01 c |
| T-20 | 3.91 ± 0.94 a | 16.87 ± 1.86 a | 0.09 ± 0.01 a | 0.12 ± 0.01 ab | 0.24 ± 0.01 a | 0.23 ± 0.02 a |
| T-30 | 3.79 ± 0.69 ab | 12.92 ± 3.25 ab | 0.11 ± 0.01 a | 0.13 ± 0.008 a | 0.25 ± 0.01 a | 0.23 ± 0.02 a |
| T-40 | 2.41 ± 0.48 b | 10.65 ± 1.85 b | 0.08 ± 0.007 b | 0.09 ± 0.007 b | 0.16 ± 0.009 b | 0.21 ± 0.02 ab |
| T-50 | 3.07 ± 0.48 ab | 10.61 ± 2.45 b | 0.12 ± 0.009 a | 0.12 ± 0.01 ab | 0.19 ± 0.02 ab | 0.16 ± 0.02 bc |

Note: Lowercase letters indicate significant differences between different treatments ($p < 0.05$).

*3.4. Effects on Fruit Quality*

Tables 6 and 7 show that different interstock lengths significantly affected fruit quality in both 2020 and 2021. In 2020, the single fruit weight of T-30 (269.89 g) was significantly higher than that of T-10. There was no significant difference in the fruit shape index among these treatments. In the following year, there were significant differences in single fruit weight, fruit shape index, and fruit skin color. The T-20 single fruit weight was the highest (284.82 g), and was significantly higher than the T-10. Additionally, the fruit shape index of T-20 was the highest at 0.85, which was significantly higher than that of T-30 and T-40. The evaluation of fruit skin color showed that the fruit brightness L* of T-30 was 47.55, which was significantly higher than T-20, T-40, and T-50. T-30 had the highest a* (red–green color) value (29.21) of fruit skin, which was significantly higher than T-20. T-20 had the highest b* (yellow–blue color) value (16.58) of fruit skin, which was significantly higher than other treatments.

**Table 6.** Effects of interstock length on fruit quality (2020).

| Treatment | Single Fruit Weight (g) | Vertical Diameter (mm) | Transverse Diameter (mm) | Fruit Shape Index | Total Soluble Solids Content(%) | Flesh Firmness (kg/cm$^2$) | Malic Acid Content (%) | Ratio of Soluble Solid to Acid |
|---|---|---|---|---|---|---|---|---|
| T-10 | 237.56 ± 42.65 b | 65.57 ± 6.31 b | 80.87 ± 4.81 b | 0.81 ± 0.07 a | 14.79 ± 1.60 a | 8.31 ± 0.96 a | 0.41 ± 0.06 ab | 36.31 ± 4.56 b |
| T-20 | 262.13 ± 35.55 ab | 69.10 ± 5.78 a | 85.09 ± 4.41 a | 0.81 ± 0.05 a | 14.75 ± 1.25 a | 7.99 ± 0.64 ab | 0.43 ± 0.14 a | 36.76 ± 9.40 b |
| T-30 | 269.89 ± 41.75 a | 69.73 ± 4.67 a | 85.47 ± 2.48 a | 0.82 ± 0.05 a | 14.95 ± 0.47 a | 8.47 ± 0.85 a | 0.34 ± 0.06 c | 43.70 ± 6.09 a |
| T-40 | 259.75 ± 41.33 ab | 69.16 ± 5.38 a | 84.17 ± 4.38 a | 0.82 ± 0.07 a | 14.38 ± 0.91 a | 7.98 ± 0.99 ab | 0.36 ± 0.06 bc | 40.48 ± 6.69 ab |
| T-50 | 257.34 ± 44.41 ab | 67.34 ± 4.14 ab | 83.89 ± 5.01 ab | 0.80 ± 0.05 a | 14.19 ± 1.68 a | 7.47 ± 0.79 b | 0.44 ± 0.12 a | 36.45 ± 10.08 b |

Note: Lowercase letters indicate significant differences between different treatments ($p < 0.05$).

The 2020 and 2021 results showed that different interstock length significantly affected the internal quality of the fruits. There were significant differences in flesh firmness, malic acid content, and the soluble solids: acid ratio among the different treatments. As shown in Tables 6 and 7, in 2020, the flesh firmness of T-30 was significantly higher than T-50, and the malic acid content was 0.34%, which was significantly lower than that of T-10, T-20, and T-50.The soluble solids: acid ratio of T-30 (43.7) was significantly higher than that of T-10, T-20, and T-50, and there was no significant difference in the content of total soluble solids among different treatments. In 2021, the flesh firmness of T-20 was the highest, and significantly higher than T-10 and T-40. The malic acid content of T-30 fruits was significantly lower than the other treatments. The T-30 fruits had the highest soluble solid: acid ratio (55.51), which was significantly higher than T-40. There was no significant difference in total soluble solids content among the treatments.

**Table 7.** Effects of interstock length on fruit quality (2021).

| Treatment | Single Fruit Weight (g) | Vertical Diameter (mm) | Transverse Diameter (mm) | Fruit Shape Index | Index of Peel Colors | | | Total Soluble Solids Content (%) | Flesh Firmness (kg/cm$^2$) | Malic Acid Content (%) | Ratio of Soluble Solid to Acid |
|---|---|---|---|---|---|---|---|---|---|---|---|
| | | | | | L* | a* | b* | | | | |
| T-10 | 243.38 ± 37.22 b | 68.43 ± 2.60 b | 82.77 ± 1.97 a | 0.83 ± 0.02 ab | 44.72 ± 4.05 ab | 27.67 ± 3.66 ab | 11.57 ± 1.55 b | 15.33 ± 1.27 a | 7.71 ± 0.46 b | 0.35 ± 0.08 a | 46.21 ± 10.91 ab |
| T-20 | 284.82 ± 41.99 a | 73.08 ± 1.76 a | 86.04 ± 1.85 a | 0.85 ± 0.01 a | 35.44 ± 2.22 c | 25.33 ± 8.56 b | 16.58 ± 1.48 a | 15.47 ± 1.50 a | 9.42 ± 0.65 a | 0.35 ± 0.11 a | 46.31 ± 11.21 ab |
| T-30 | 276.05 ± 28.91 a | 69.49 ± 3.50 ab | 86.16 ± 2.22 a | 0.81 ± 0.03 b | 47.55 ± 4.53 a | 29.21 ± 4.71 a | 13.10 ± 1.89 b | 14.79 ± 1.21 a | 9.13 ± 0.42 a | 0.29 ± 0.09 b | 55.51 ± 16.30 a⁻ |
| T-40 | 275.17 ± 30.21 a | 67.69 ± 0.88 b | 83.91 ± 3.56 a | 0.81 ± 0.01 b | 43.41 ± 2.85 b | 27.51 ± 4.96 ab | 9.73b ± 2.42 b | 14.77 ± 1.19 a | 7.59 ± 1.67 b | 0.34 ± 0.07 a | 44.48 ± 7.97 b |
| T-50 | 276.06 ± 56.32 a | 71.58 ± 3.02 ab | 85.75 ± 2.26 a | 0.84 ± 0.03 ab | 42.22 ± 2.51 b | 26.69 ± 2.17 ab | 10.12 ± 1.27 b | 14.58 ± 1.31 a | 8.75 ± 0.88 a | 0.33 ± 0.11 a | 47.81 ± 12.56 ab |

Note: Lowercase letters indicate significant differences between different treatments ($p < 0.05$). L* represented the color brightness, a* represented the peel's red-green value, and b* represented the yellow-blue value.

### 3.5. Effects of Dwarfing Interstock Length on Root Growth

3.5.1. Effect on Root Length Density

As shown in Figure 1A, the root length density of each treatment generally increased firstly, then decreased, and then increased again. From mid-March to mid-June, root length density increased, and the growth peak of roots occurred in June. The root length density of T-30 (3.52 mm·cm$^{-3}$) was significantly higher than that of T-50. The root growth rate of each treatment decreased continually from mid-June to mid-August. The growth rate of roots was slowest in August. At this time, the root length density (0.97 mm·cm$^{-3}$) of T-50 was significantly lower than that of the other treatments. The root growth rate increased slightly from mid-August to mid-September.

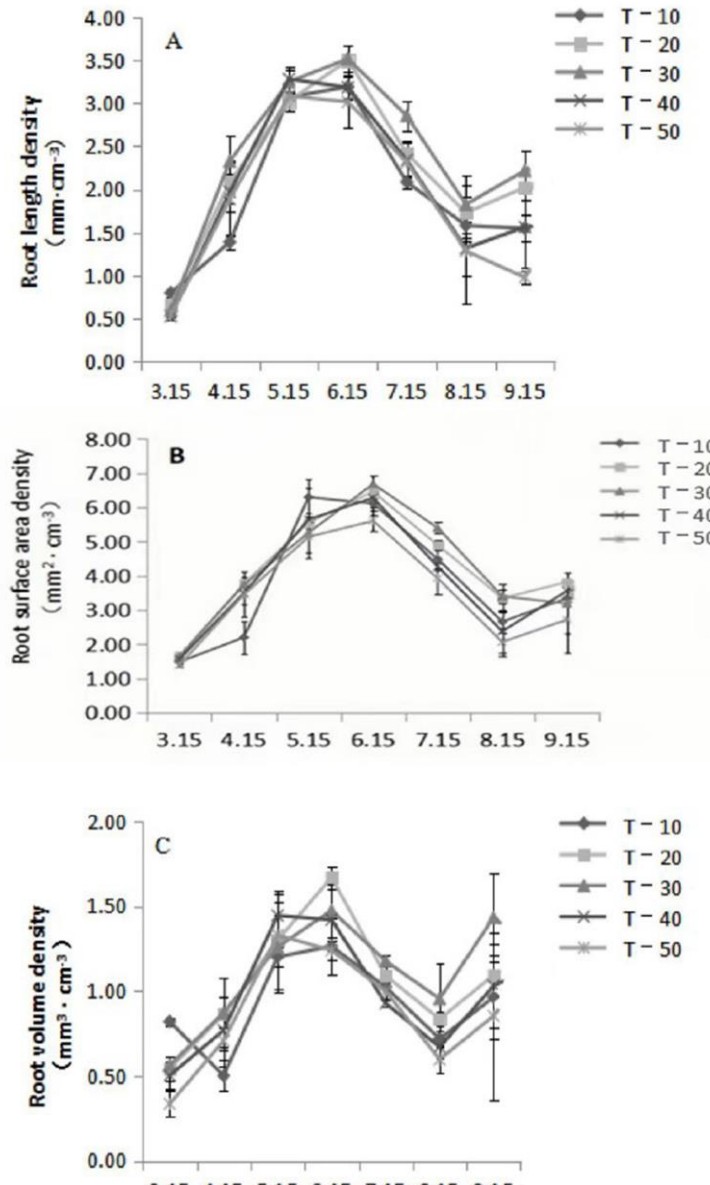

**Figure 1.** Effects of interstock length on root growth. (**A**)—root length density; (**B**)—root surface area density; (**C**)—root volume density.

3.5.2. Effect on Root Surface Density

As shown in Figure 1B, the root surface density of each treatment generally was bimodal with an increasing–decreasing–increasing pattern. The root surface density increased from mid-March to mid-June, and the root growth peak occurred in June. At

thatmoment, the root surface area density of T-30 (6.68 mm$^2 \cdot$cm$^{-3}$) was significantly higher than that of T-10 and T-50. The root growth of each treatment continually decreased from mid-June to mid-August. The root growth rate was the lowest in August. At that moment, the root surface area density of the T-50 treatment (2.06 mm$^2 \cdot$cm$^{-3}$) was significantly higher than that of T-20 and T-30, and the growth rate of the roots increased from mid-August to mid-September.

### 3.5.3. Effect on Root Volume Density

As shown in Figure 1C, the root volume density of each treatment generally showed a double-peak curve with the same trend (increasing–decreasing–increasing) determined for root length and surface density. From mid-March to mid-June, root volume density increased, peaking in June. At this time, the root volume density of the T-20 treatment was higher than that of T-10 and T-50. The root growth of each treatment continually decreased from mid-June to mid-August. The growth rate of the roots was the lowest in August. At that moment, the root volume density (0.95 mm$^3 \cdot$cm$^{-3}$) of T-30 was significantly higher than that of T-10, T-40, and T-50. From mid-August to mid-September, the growth of roots increased slightly.

As previously stated results showed, there were significant differences in rootgrowth of apple trees with different interstock lengths. At the peak of root growth in summer (June), the root length density, root surface density, and root volume density decreased with interstock length. When the interstock length was 50 cm, root growth was inhibited compared with other treatments. When the interstock length was 30 cm, the root length density, root surface density, and root volume density were significantly higher than the other treatments.

### 3.5.4. Effect on Root Length Density of Dead Roots

As shown in Figure 2, the root length density of dead roots in each treatment generally showed a single-peak curve with an increasing then decreasing trend. From mid-April to mid-July, the root length density of dead roots increased, and the death peak of roots occurred in July. At that time, the root length density of the dead roots (0.38 mm$\cdot$cm$^{-3}$) of T-50 was significantly higher than that of T-10, T-20, and T-40. The root length density of the dead roots showed a downward trend from mid-July to mid-September, indicating that the rate of root death of each treatment decreased continuously from mid-July to mid-September.

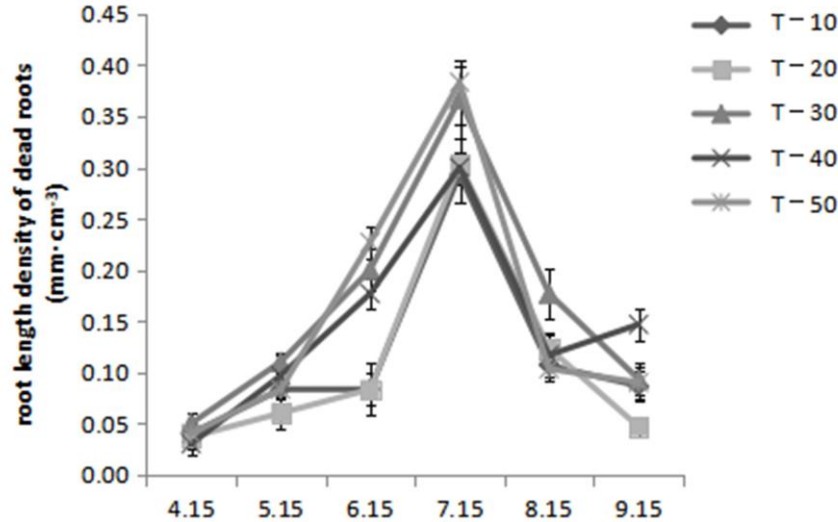

**Figure 2.** Effect of dwarfing interstock length on root length density of dead roots.

### 3.5.5. Effect on Root Turnover Rate

Figure 3 shows the significant differences in root turnover rate between treatments. The root turnover rate increased with interstock length. The T-50 root turnover rate was the highest (0.56), which was significantly higher than other treatments, followed by T-40 (0.47), and T-20 (0.34).

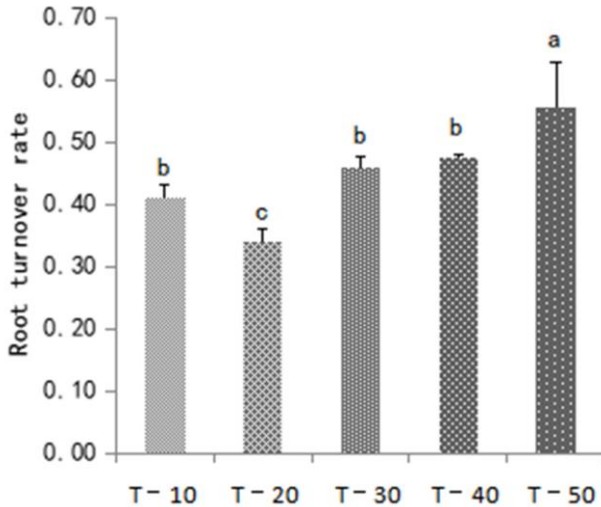

**Figure 3.** Effect of dwarfing interstock length on root turnover rate. Lowercase letters indicate significant differences between treatments ($p < 0.05$).

## 4. Discussion

### 4.1. Effect of Dwarfing Interstock Length on the Growth of the Aboveground Part of Apple Trees

Dwarfing interstock length affects the movement of water, mineral elements, and other nutrients [19]. As interstock length increases, the distance between the canopy and roots is longer, which decreases the rate of water and nutrients transport, restricts the growth of the aboveground part, and decreases tree vigor [11].The greater the interstock length, the greater the restricting effect on shoot growth; therefore, the better the dwarfing effect [20]. Our study also found that different interstock lengths significantly affected the size of apple trees, as vegetative growth decreased with the increase in dwarfing interstock length. Similar results were obtained on rootstocks, such as GM256, M9, and MM106 [6,13], showing that interstock length can be increased to increase the dwarfing effect. The vigor reduction in high grafted materials could be a useful tool in high-density plantations that require low vigor plants [21], which are best suited to obtaining high yields [11].

A well-branched maiden tree is a key factor for an early crop in the orchard. The number of lateral branches provides the opportunity to obtain good tree architecture. The number of lateral shoots was significantly higher when using the 30 cm interstock compared with 15 cm interstock, and the lateral shoot length and diameter were also significantly affected by interstock length, while interstock length also significantly affected lateral shoot length and diameter [6]. In addition, good lateral branch growth condition provides earlier and higher yields [22]. The key to a high and stable yield of fruit trees lies in the appropriate quantity of branches and the ratio of different kinds of branches. Different interstocks significantly affect the share of cluster branches and short branches on apple trees [23]. This study showed that with the increase in interstock length, the proportion of short branches increased and then decreased. When the interstock length was 30 cm, the proportion of short branches was the highest, which helped control the growth of the trees; making it possible for early flowering and bearing of fruit, which could inhibit tree growth and greatly increase the high-yield potential. The appropriate dwarfing interstock length could keep trees growing and effectively improve the high-yield potential to achieve high and stable yields.

### 4.2. Effects of Dwarfing Interstock Length on Yield and Fruit Quality of Apple Trees

With the rapidly developing Chinese economy, consumers are paying more attention to fruit quality, and apple fruit production is being transformed from high yield to high quality. The age at which trees begin to bear fruit and the cumulative yield are significantly affected by the interstock [6,11]. Early fruit production in double-grafted trees might result from the interaction between vegetative and reproductive growth [6], because reproductive growth is stimulated by reduced vegetative growth [24–26].Moreover, increasing interstock length increased yield per tree [6]. The present study investigated the effect of dwarfing interstock length on yield per apple tree. Our results showed that when the interstock length of Jizhen 2 was 30cm, the yield per tree and the yield per unit cross-sectional area were the highest.

The interstock (variety, planting depth, or length) also has an influence on fruit quality. A survey of 3-year-old plants of the apple cultivar 'Changfu 2' grafted on M26 dwarfing interstock at different planting depths (5, 10, 15, 20, and 30 cm), indicated that, in the 15-cm treatment, the fruit yield was higher, the single fruit weight was greater, and the fruit quality was improved in terms of soluble solids content, firmness, and color [27]. Another study indicated that apples (cv. Gala) with greater firmness were observed as the length of the interstock (EM-9) was increased (10, 15, 20, 25, and 30 cm) [12]. The interstock also affected the total fruit acid content [23]. We also studied the effects of different dwarfing interstock lengths on fruit quality. The results showed that when the interstock length of Jizhen 2 was 30 cm, the single fruit weight was the highest, firmness was higher, malic acid content was the lowest, the solublesolids: acid ratio was the highest, and brightness L* and red-green value a* of the fruit skin were the highest.

Nutrients transported by the roots are primarily used for tree growth. When the interstock length is too short, nutrients obtained for reproduction and growth are reduced, resulting in low fruit yield and poor fruit quality. When the dwarfing interstock length exceeds a certain range, water and nutrients transported upward by the roots are consumed excessively, and the nutrients obtained from the aboveground parts of the tree are limited, which greatly inhibits the growth of the tree [10], reduces the physiological activity of leaves, weakens the photosynthetic rate, reduces the accumulation of dry matter, also results in poor fruit yield and quality. Appropriate interstock length can enhance the physiological activity of leaves by increasing the proportion of short branches, which enhances the photosynthetic characteristics of fruit tree. The increased photosynthesis increase dry matter accumulation and promotes the transportation of photosynthetic products to the fruits, thus improving early blossoming and early fruit–setting, which directly enhances fruit quality.

### 4.3. Effects of Dwarfing Interstock Length on Root Growth

Stable root function depends on the amount of roots, their mortality, and the frequency of root turnover, which can be influenced by internal and external conditions [28]. The growth peak of the roots was different from that of new shoots. In this study, the growth peaks of the roots occurred in mid-June and mid-September, when the spring shoots and autumn shoots were at the period of slow growth and the period of stopped growth, respectively. During the observation period, the growth of the roots in each treatment showed a bimodal-shaped curve of an increasing–decreasing–increasing pattern. The root length density, root surface density, and root volume density increased after March, and peaked in mid-June (slow growth to the stopped state period of spring shoots). The root growth in each treatment was inhibited in mid-August. The root length density, root surface density, and root volume density all decreased, and root growth was inhibited. In mid-September (slow growth period of autumn shoots), the growth of roots rose slightly. At this time, the demand for nutrients by the aboveground parts decreased, and most of the nutrients in the fruit tree returned to the roots, forming the second growth peak. However, the death peak of fine roots coincided with the vigorous growth period of the autumn shoots. In this study, root mortality was highest in mid-July, which may be related to the

high soil temperatures or excessive rainfall during the summer, when the roots were in a water-saturated state, leading to hypoxia.

The quantitative distribution of roots in the soil differed with different interstock lengths. The roots remained stable, allowing them to be fully functional. The distribution range and depth of root affected the absorption of water, mineral elements, and other nutrients by trees, and had an important impact on fruit tree yield [29]. The dwarfing interstock length directly affected the growth of the fruit tree roots. The root growth decreased with the increase in dwarfing interstock length. The increase in dead root length density inhibited the root's nutrient absorption efficiency. If the interstock is too long, the roots' absorption and secretion functions will be weakened, indirectly affecting the tree's growth, increasing the root turnover frequency, and enhancing the dwarfing effect. When the dwarfing interstock length was 30cm, the source–sink developed harmoniously, and the growth of the roots was better than in the other treatments, which further promoted photo synthesis of the aboveground parts of the tree, increased nutrient accumulation and fruit yield, and improved fruit quality. Good growth of the aboveground parts of the tree, in turn, would promote root growth and provide advantageous conditions for the roots to absorb nutrients.

## 5. Conclusions

The dwarfing degree of trees gradually increased as the dwarfing interstock length was increased. T-30 had the fewest long branches, the most short branches, the highest proportion of short branches, and the highest fruit–set rate of inflorescences and a single flowers. Overall, the fruit quality of the T–30 treatment was the best (the highest single fruit weight, the highest soluble solids: acid ratio, the highest color brightness, and better coloration of the red fruit skin). Combining with these results, 30 cm was the most suitable length for the Tianhong 2 apple trees when the Jizhen 2 was used as the interstock, compared with 10, 20, 40, and 50 cm.

Apple dwarf intensive cultivation has become the mainstream model of the modern apple industry throughout the world. The use of dwarf interstock to control tree growth is an important way to develop apple dwarf intensive cultivation in China. Different dwarfing interstocks and different grafting lengths of the same dwarfing interstock have great effects on the growth and fruit of apple trees. Jizhen 2 is widely used in the Bohai Bay producing area and parts of the Loess Plateau producing area as apple interstock. In these areas, we recommend an application length of 30 cm for this kind of interstock according to our research result. For other types of apple interstocks or interstock of other fruit trees, the application length remains to be studied.

**Author Contributions:** Data curation, B.Y.; Funding acquisition, J.X.; Investigation, Z.S.; Methodology, X.Z. and J.X.; Resources, B.L. and Z.L.; Writing—original draft, S.Z. All authors have read and agreed to the published version of the manuscript.

**Funding:** This research was funded by the China Agriculture Research System (CARS-27), the Project of Science and Technology Department of Hebei Province (16226312D-7), the Natural Science Foundation of China (32002008), the Natural Science Foundation of Hebei Province (C2020204015), the Introduced Talents Scientific Research Project of Hebei Agricultural University (ZD201706), and the Key Research and Development Program of Hebei Province (20326802D).

**Data Availability Statement:** Data sharing not applicable; No new data were created or analyzed in this study. Data sharing is not applicable to this article.

**Acknowledgments:** We thank Luqiang Yang for the management of the apple trees.

**Conflicts of Interest:** The authors declare no conflict of interest or personal relationships that could have appeared to influence the work reported in this paper.

## Abbreviations

TCSA, unit cross-sectional area; TSS, total soluble solids.

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
