# Peer review of "Effects of Dwarfing Interstock Length on the Growth and Fruit of Apple Tree"

_horticulturae, doi:10.3390/horticulturae9010040_

Round 1

Reviewer 1 Report (Previous Reviewer 3)

I suggest a title change to Effects of Dwarfing Interstock on apple performances. 

Revise this sentence: Apple dwarfing and dense planting has the advantages of dwarfing...

English might still be improved. 

In the conclusion section please provide the broader application of your results. Have in mind the wide audience of the Horticulturae Journal and recommend the application of investigated interstocks or suggest similar investigations in other germplasm and other ecological conditions. 

Author Response

Thanks very much for your professional comments and precious time.

Point 1: I suggest a title change to Effects of Dwarfing Interstock on apple performances. 

Response 1: Thanks for your valuable comment. The title ‘Effects of Dwarfing Interstock on apple performances’ is more concise. But our research focus is ‘Effects of different Dwarfing Interstock length on the Growth and Fruit of Apple Tree’.  Take into account your suggestions while highlighting our research point, we decide to change the title to ‘Effects of Dwarfing Interstock Length on the Growth and Fruit of Apple Tree’.

Point 2: Revise this sentence: Apple dwarfing and dense planting has the advantages of dwarfing...

Response 2: Thanks for your valuable comment. We have deleted the second ‘dwarfing’ and revised this sentence to ‘Apple dwarfing and dense planting has the advantages of early blooming and fruiting, high yield, and good quality, facilitating intensive cultivation management’.

Point 3: English might still be improved.

Response 3: Thanks for your valuable comment. We have requested one our native English-speaking colleague to check our manuscript to improve the English. All changes have been marked up using the “Track Changes” function in the revised manuscript.

Point 4: In the conclusion section please provide the broader application of your results. Have in mind the wide audience of the Horticulturae Journal and recommend the application of investigated interstocks or suggest similar investigations in other germplasm and other ecological conditions. 

Response 4: Thanks for your helpful suggestion. This view is very helpful to improve  this paper. We have provided the broader application of our results, recommend the application of investigated interstocks, and suggested similar investigations in other germplasm and other ecological conditions. These contents are listed at the second paragraph of the conclusion.

Reviewer 2 Report (Previous Reviewer 2)

I reviewed the revised version of manuscript, authors responded to all my concerns and revised manuscript accordingly. I suggest this revised version can be accepted for publication.

Best regards

Author Response

Thank you very much for your approval of our manuscript! Best wishes!

Reviewer 3 Report (Previous Reviewer 1)

Thank you for your new information and references. Paper, for me, is very clear and well written. References are correct and richer than the last version the paper.

Author Response

Thank you very much for your approval of our manuscript! Best wishes!

This manuscript is a resubmission of an earlier submission. The following is a list of the peer review reports and author responses from that submission.

Round 1

Reviewer 1 Report

I have read the manuscript Effects of Dwarfing Interstock Length on the Growth of Overground Part and Root and Fruit Quality of Apple written by Shasha Zhou et al, for publication of Horticulturae MDPI. In this study the authors have investigated the effects of Dwarfing interstock lenght on the growth of roots, oveground parts and fruit quality of twenty-five apple trees, for two years. They have found significant differences in tree height (above all in the second year of studies), in stem girth of interstock and scion (both years), crown diameter, number of short branches, ratio of short branches, fruit setting rate of a single flower, yield per tree and per unit cross-sectional area, single fruit weight, ratio of soluble solids to acid, color brightness, red-green value of the fruit skin, root length density, root surface and volume density. The overall research is well conducted, the study is very interesting for readers and the manuscript is much valuable. I found some points, especially in the introduction and materials and methods, to better explain, with other references, the meaning of some concepts.

Abstract: Concepts related to fruit quality and roots are well listed and discussed, those related to the aerial part of trees are less discussed. You have obtained several and significant differences in the overground part of your trees. Try to listed other differences in the abstract and not only the dwarfing degree, in general.

Introduction: Ln 38-39. Please, cite some reference.

Ln 42-43. You cited five studies (from number 8 to number 12 reference) about the effects of dwarfing interstocks, but just one relates to the apple trees(Di Vaio, C.; Cirillo, C.; Buccheri, M.; Limongelli, F. Effect of interstock (M. 9 and M. 27) on vegetative growth and yield of apple trees 446 ("v. “Annurca”). Sci. Hortic. 2009, 119 (3), 270–274.). Three of them relate to peach trees and the last one to mango trees. Please, cite other publications regarding apple trees.

Materials and Methods: Do you have any information regarding soil type?

Reviewer 2 Report

Comments to authors,

The manuscript entitled 'Effects of Dwarfing Interstock Length on the Growth of Over-2ground Part and Root and Fruit Quality of Apple' aims to evaluate the effect of interstock length on canopy and fruit parameters. Introduction part is focused on the paper, methodologies are given in detail, and results presentation, interpretation and discussion is satisfactory. In the text, is a number of other points that need attention (see some comments to the authors).

Comments to authors:

Abstract:

Line 21:  highest color brightness (L*), and better red skin coloration (higher a*).

Material and methods:

Line 130: firmness instead of hardness

Results:

Line 190: I suggest: T-30 had significantly lower number of long branches as compared to other  treatments.

Line 233: highest parameter a* (red-green color) of fruit skin

Line 234: highest parameter b* (yellow-blue color) of fruit skin

Discussion:

Line 326: higher instead of increased

Discussion:

4.2. Effect of dwarfing interstock length on yield and fruit quality of apple trees

I suggest to comment more on firmness,  acid, ratio TSS/acid using also literature data.

Conclusions

I suggest to mention single fruit quality parameters as influenced by interstock length and add future prospects.

Reviewer 3 Report

In general, the manuscript seems to be unfinished and some mistakes are likely the result of the mistakenly submitted document. Regardless, comments and suggestions are given below. 

Lines 4-15

Information on authors' names and affiliations, as well as citation details, must be corrected in accordance with journal's template. Please delete unnecessary information added to the list of authors' names.

ABSTRACT

Lines 16-26

Please restructure the abstract in order to follow the journal's recommendation (template), by adding more background and methods information, and not only the main findings of the paper.

Line 29

Check the font used in this line.

Lines 30-31

Use (,) instead of (:)

TCSA, unit trunk cross-sectional area;

TSS, total soluble solids.

INTRODUCTION

Line 33-35

Apple Dwarfing and dense planting technologies have the advantages of dwarfing, early blooming and fruiting, high yield, and good fruit quality, facilitating intensive cultivation management [1], which has become the mainstream and direction of modern applefruit production globally.

Line 36

Dwarfing interstock is one dwarfing and dense planting such technology

Line 45

Change: increase or enhance rather than increment tree density

Lines 45-48

Previous Some studies have shown showed that the dwarfing effect on apple trees increased with the interstock length [5,13]. Some studies showed, while other authors found that different interstock lengths significantly affected the fruit quality and yield of the 'Royal Gala' and 'Nagafu No.2' apples [14-15].

Lines 50-51

The sentence ‘The root is the foundation of the growth of fruit trees, and it has desludging, regulation, storage, synthesis, metabolism, and absorption functions.’ is very similar to the sentence used in the discussion chapter lines 370-371

Rewrite those sentences in both chapters. What do you mean by desludging function?

Line 60

Change: variant to cultivar or variety

Line 69

Be more specific in this sentence: ‘Currently, there is no report on the interstock length of Jizhen 2.’

Did you mean there is no report on the effect of different interstock length of Jizhen 2 on the tree vegetatitive growth, yielding and/or fruit traits?

 Lines 70-75

The aim must be more clearly stated, these two sentences are almost the same regarding the information they provide, please be careful to not repeat yourself throughout the paper. Better use term ‘aboveground plant parts’ than ‘aboveground tissues’, since you were investigated interstock effects on vegetative and generative parameters but not on histological level. Since this is the introduction chapter it is more appropriate to say ‘this work aims to elucidate’ than ‘this work elucidates the effects…’.

MATERIALS AND METHODS

Lines 77-78

Delete the following words since you stated the aim in the previous section: ‘The research to study the influence of different interstock lengths on the growth of apple trees was conducted at the Nanshennan Apple Experimental Base, Shunping County, Hebei Province (38°59′ north latitude, 114°54′ east longitude).’

Lines 81-84

I suggest to divide and simplify this sentence, in order to be more readable. ‘The rootstock was Malus × robusta Rehder annual seedlings, the interstock branches (Jizhen 2) of different lengths were grafted on the rootstock, and the Tianhong 2 apple buds were grafted on the interstocks in 2017.’

For example: ‘Malus × robusta Rehder annual seedlings were used as rootstocks, while Jizhen 2 was used as an interstock. The Tianhong 2 apple buds were grafted on the interstocks of different lengths in 2017.’

Subsections 2.1. to 2.3.

Please explain why some parameters were measured and evaluated only in one year? For example, why branch types assessment was carried out only in 2020, and not 2019? Also, why did you choose to investigate fruit set rates only in 2021, while yield and fruit quality were assessed in both 2020 and 2021?

Line 93

Rewrite: ‘When the apple trees stopped growing in October 2019 and October 2020…’ to ‘After the vegetative growth has stopped, in October 2019-2020, …’

Line 99

I suggest to use ‘the share of different branches in the total number of branches (expressed in percentages)’ better than ‘ratio of different branches’. Please uniform all terms used in methods and later in results and discussion sections, including tables.

Line 103

The title must point to all the subsection content, so the title ‘Determination of fruit set and yield’ is more appropriate.

Line 113

Instead of ‘we counted’ use listed parameters ‘were measured

Lines 115-120

This sentence is too long and confusing. When you explain two types of yield that you have assessed, in the next sentence explain how you calculated yield according to interstock and scion TCSA. In that way you will avoid so many brackets in one sentence. Also, for all parameters in the paper write measure units in the brackets, in this case, express yield in kg, both per plant and per unit of TCSA. In accordance with that, change the values from grams to kg in table 5.

Line 122

Change: ‘Fifty uniform fruits (10 fruits per tree), harvested/picked at the height of 1.4-1.6 m from the ground above the ground around the and from all sides of the canopy…’

Line 124

Better say ‘parameters’ than ‘indexes

Line 130

Decide do you use ‘fruit hardness’ or ‘flesh firmness’ and uniform it through the text and tables.

Line 141

More correct is to say that measurement was carried out on the all four sides of the fruit than that these sides were measured.

Line 142

You already said that 50 fruits were measured, so in this place there is no need to repeat that information.

Line 146

Change: ‘of the root of apple trees’ to ‘of the apple trees’ roots

Line 154

Check the unit, did you mean ‘The root length density (mm·cm−3)’ rather than ‘(mm2·cm−3)

Also, check and correct the units in Figure 1.

Further, check if you need to add mm in the brackets after ‘L is the total root length of a single minirhizotron‘ in the same line.

RESULTS

Line 166

Add word vegetative in the title: ‘3.1. Effect of dwarfing interstock length on the vegetative growth of apple trees’

Lines 168-180

I suggest to analyze first all parameters from 2019, and then all from 2020, with the more accentuated beginning in line 173 where you can start with ‘In the next year…’. For example, in lines 176-177 only 2019 data was analyzed, so those shifts from 2019 to 2020 data is a bit confusing. Why the crown diameter values are missing in the table 1? Also, you can add some small changes to make text more readable, such as: line 172 You can combine those two sentences in this way: ‘The T-10 interstock stem girth was significantly higher than other treatments, while the T-10 scion stem girth was significantly higher than the T-40 and T-50.’ line 173 change: ‘In the next year, the T-10 tree height in 2020 was 378.00 cm, which was significantly higher than with T-40 and T-50.’

Please check fonts in all tables and use font according to template.

Also, where is missing, add brackets for units in tables.

In all tables, add according to which statistical test the significant differences were determined.

All tables must be self-explainable.

Line 193

You could state that: ‘Because share of branches belonging to different types follows the number of those branches, the proportion percentage of long branches in T-30 treatment was significantly lower than in T-10, T-40, and T-50.’ Please combine the next two sentences with the previous explanation of number of branches because sentences ‘There was no significant difference in the number of medium branches among treatments.’ and ‘There was no significant difference in the proportion of medium branches among different treatments.’ are practically the same.

I suggest the following change in Table 3. title ‘Effects of different dwarfing interstock lengths on number and share of branches belonging to different types the kind of branches (2020)’.

Also, I suggest the following change in Table 4. title ‘Effects of dwarfing interstock lengths on flowering, and fruit-setting rates (2021).’

Line 204

The value of 34% for T-30 treatment in the text is different in the table were the value is 28%. Please check your data.

Line 213

Change: ‘the yield per cm2 TCSA of T-50 interstock was the highest at reaching 0.12 kg/cm2

Line 217

Change: ‘was the highest with at 0.25 kg/cm2

Lines 218-219

Change: ‘The yield per TCSA of the T-30 scion was the highest, with 0.23 kg/cm2, which was significantly higher than T-10 and T-50.’

In Table 5. title add year of investigation (2021), in order to be uniform with other tables.

Change subtitle 3.4. Effect on Fruit Quality font to italic.

Line 224

Please delete duplicate of subtitle.

Line 225-226

Change: ‘Tables 6 and 7 showed that different lengths of interstock significantly influenced fruit quality in both 2020 and 2021’

Line 228-229

Change: ‘In the following year, there were significant differences in single fruit weight, fruit shape index, and fruit skin color in 2021.’

Lines 230-231

Change: ‘Also, the fruit shape index of T-20 was characterized with the highest fruit shape index of with 0.85, which was significantly higher than T-30 and T-40.’

Tables 6 and 7

In these tables values for ratio of soluble solid to acid were showed, but explanation of this parameter in methods section is missing. Please provide that information in addition to described method of TSS data acquisition which was measured in order to calculate mentioned ratio.

Lines 248-250

In this sentence, you commented content of soluble solids, although the data for this parameter was not shown. Please stay within the same parameter which is analyzed in the manuscript, that is, ratio of soluble solid to acid which you need to explain in methods related to measured TSS.

Please change subtitle 3.5. to ‘Effect of dwarfing interstock length on the root growth of root’

Lines 269-270

Change: ‘As shown in Figure 1-B, the root surface density of each treatment generally showed a bimodal curve with trend of increasing-decreasing-increasing then decreasing and then increasing trend.’

Lines 278-279

Change: ‘As shown in Figure 1-C, the root volume density of each treatment generally showed a double-peak curve with the same trend determined for root length and surface density a t of increasing - decreasing - increasing trend.’

Line 283

Use ‘At that moment’ instead of ‘At this time

Line 284

Correct unit in this sentence.

Line 286

Add ‘As previously stated results showed, there were…’

Lines 297-299

Can you explain better what did you mean by the second part of the sentence, it is not very clear? ‘Each treatment's root death decreased continuously from mid-July to mid-September, and root length density of dead roots decreased this time.’

DISCUSSION

In general, this section must be rewritten, the text is hard to follow and some references are missing.  Some suggestions and comments are listed below:

Line 313

Delete: ‘tree body’

Think about moving sentence from lines 319-321 to line 313, in front of ‘The greater the interstock length…’

Line 315-317

Change: ‘Our study also found that different interstock lengths significantly affected the size of apple trees, and the growth of apple trees whereby the vegetative growth decreased with the increasing of dwarfing interstock length.’

Lines 326-328

Change: ‘The number of lateral shoots was significantly increasesd with 30 cm interstock length compared with 15 cm interstock length, and the lateral shoot length and diameter were also significantly affected by interstock length while interstock length also significantly affects lateral shoot length and diameter [5].’

Line 331

What did you mean by tufted branches?

Line 343

Some reference is missing.

Lines 346-353

Paragraph change: ‘Moreover, the increase in the length of interstock increasesd yield per tree [5]. This research studied the influence of dwarfing interstock length on yield per apple tree v. 347 ‘Tianhog 2’. The results showed Our results showed that when the interstock length of Jizhen 2 was 30 cm, the yield per tree and the yield per unit cross-sectional area were the highest. We also studied the effects of different dwarfing interstock lengths on fruit quality. The results showed that when the interstock length of Jizhen 2 was 30 cm,Within the same treatment, the single fruit weight was the highest, the ratio of soluble solids to acid was the highest, and the brightness L* and red- green value a* of the fruit skin was were the highest.’

Line 354

I suggest deleting the sentence ‘Does the interstock length of dwarf crops affect the transport of nutrients?’ as that statement is already written.

Lines 359 and 364

Please rewrite these sentences without repeating the same words in the same paragraph: ‘resulting in low fruit yield and poor fruit quality’ and ‘resulting in poor fruit yield and quality’.

Line 368

Change: ‘…and then directly improving fruit quality’ to ‘…which directly enhance fruit quality’.

Lines 370-371

Avoid using the same or very similar sentences throughout the manuscript, I suggest deleting this sentence since in the introduction is the similar one.

Line 372

What did you imply by standing stock?

Line 377

Some reference is missing.

Line 377-378

Delete the sentence due to repeating: ‘The growth dynamics of the root showed a bimodal-shaped curve.’

Lines 388-390

Change: ‘In this study, rRoot mortality was the highestconcentrated in mid-July, which may be related to high soil temperatures or excessive rainfall in summer, when the roots were in a water-saturated state, leading to hypoxia.’ Did you have some meteorological data to cite, with the measured parameters in that year which are in accordance with your assumption?

Lines 397-402

Two almost the same sentences one after another, please delete one in the context of that paragraph.

CONSLUSIONS

Please do not use adjectives such as good, better or the best treatment without explaining why, both in the discussion and in conclusion. After finishing discussion, please rewrite the conclusion highlighting the best interstock length for the achievement of the particular production goals.

REFERENCES

Please uniform all references according to the template given by the Journal.